# Comprehensive analysis of total knee arthroplasty kinematics and functional recovery: Exploring full-body gait deviations in patients with knee osteoarthritis

Xavier Gasparutto[1]*, Alice Bonnefoy-Mazure[1], Michael Attias[2], Katia Turcot[3], Stéphane Armand[1], Hermès H. Miozzari[4]

**1** Kinesiology Laboratory, Geneva University Hospitals and University of Geneva, Geneva, Switzerland, **2** HES-SO University of Applied Sciences and Arts Western Switzerland, School of Health Sciences, Geneva, Switzerland, **3** Centre for Interdisciplinary Research in Rehabilitation and Social Integration (CIRRIS), Laval University, Quebec City, Canada, **4** Division of Orthopedic Surgery and Musculoskeletal Trauma Care, Geneva University Hospitals and University of Geneva, Geneva, Switzerland

* xavier.gasparutto@hug.ch

**Data Availability Statement:** All data are available from the Yareta database (accession number:

## Abstract

Total Knee Arthroplasty has well-established success in relieving knee pain and improving function but patients do not reach functional levels of control groups after surgery and 20% of patients remain unsatisfied. To understand the different patient profiles and develop patient-specific approaches of care, functional phenotypes based on knee biomechanics during gait have been evaluated. To widen the understanding of patient's function, it seems crucial to consider the gait devieations at the whole body level. Thus, this study aims at 1) assessing the impact of knee OA on full-body gait mechanics, 2) assessing whether potential deviations persist one year after TKA surgery, and 3) their potential impact on satisfaction. To that end, clinical gait analysis was performed before and one year after surgery for 100 patients planned for unilateral primary TKA, along with 32 healthy participants as control group. Patients were clustered by applying K-means algorithms on full-body kinematic features of gait before surgery. The knee was excluded from classification to focus on full-body kinematics. Differences between groups, with controls, as well as before and after surgery were evaluated for patients reported outcome measures, kinematic features, and spatio-temporal parameters. Three functional groups were identified. One low-functioning cluster with mostly elderly women showing significant functional improvement one year after surgery, and two high-functioning clusters differentiated by pelvis tilt (anteversion vs. retroversion), sagittal knee alignment (varus vs. neutral), and knee flexion during stance phase (flexum vs. extended) that showed limited improvement one year after surgery. Satisfaction rates were similar among clusters and mental scores improved for all clusters. High functioning patients may benefit from TKA, mostly due to pain reduction, but may not see significant improvement of their function, with no clear impact on satisfaction rate. On the contrary, patients with important functional limitation are more likely to improve both pain and functional outcomes.

https://doi.org/10.26037/
yareta:4xnhbsiyqrbihlep2nst2u2hyq)

**Funding:** This study was funded by the "Fondation pour la recherche ostéoarticulaire" of Geneva (Switzerland). The funders had no role in study design, data collection and analysis, decision to publish, or preparation of the manuscript.

**Competing interests:** The authors have declared that no competing interests exist.

## 1 Introduction

Total Knee Arthroplasty (TKA), aims at alleviating the debilitating effects of knee osteoarthritis (OA), and has become increasingly prevalent in recent years [1] with well-established success in relieving knee pain [2] and improving function [1]. However, movement analysis, and more specifically clinical gait analysis, has shown that, despite significant and sustained improvements in terms of knee biomechanics [3, 4] and spatio-temporal parameters of gait [3, 5], in average patients do not reach functional levels of control groups after surgery [6].

These results provide a global understanding of the impact of the surgery on function but do not take into account the wide range of patients' functional levels before surgery. To better understand this variability and help with patient-specific approaches of care, recent studies have sought to identify functional phenotypes of knee OA based on knee biomechanics with the idea of going beyond addressing static knee alignments only [7, 8].

To characterize distinct movement phenotypes for patients with end-stage knee OA, two main approaches can be employed. The "supervised" approach relies on clinical expertise and knowledge, and considers factors such as gender, pain, body mass index, age, pre- and post-surgery PROMs (Patient-Reported Outcome Measures), knee alignment, and knee biomechanics before and after surgery [9]. In this approach patients are grouped based on features and thresholds identified from experts knowledge and subjective assessments. Conversely, the "unsupervised" approach relies on automatic classification and machine learning, and is applied to a large database with automatic feature extraction. In this approach, multidimensional data are categorized in cohesive subgroups (or clusters) based on their statistical differences and similarity [7, 8, 10]. The main interest of such a method is to identify patterns and features that may have been overlooked in large amount of data while the main limitation is the consistency between studies. Indeed, the type of clusters identified is dependent on the input data and methods used [10].

While much attention has been given to the knee's kinematics and the external knee adduction moment (KAM) in both supervised [9] and unsupervised approaches [8], it is also crucial to consider the adaptations of the whole body during movement and more specifically during gait. Indeed, patients with knee OA have been showed to present reduced gait speed and knee flexion range of motion [11] but it is not known how knee OA impacts the overall movement of the body, what adaptations patients employ to mitigate pain and maintain function, and whether such adaptations evolve or disappear with surgery. Different patient profiles might exhibit distinct gait deviations, such as limping, increased use of the trunk, or altered hip and pelvis movements, in a bid to alleviate discomfort and pain or compensate for weaknesses. Such deviations can however be detrimental in the long term. For instance, self-evaluated "limping when walking" was found to be a strong predictor of TKA revision [12], proximal gait deviations in patients after TKA were suggested as a risk factor for hip abductor muscle weakness and low-back pain [13], while the development of contralateral knee OA was associated with remaining stiff knee gait after TKA [14].

To get a better understanding of these potential gait deviations, this study undertook a thorough investigation of full-body kinematic profiles of patients with knee OA planned for primary TKA. The objective of this study were 1) assessing the impact of knee OA on full-body gait mechanics, 2) assessing whether potential gait deviations persist one year after TKA surgery, and 3) their potential impact on satisfaction. An unsupervised approach has been chosen, to determine several clusters from full-body gait kinematic and spatio-temporal parameters, while excluding knee kinematics, to focus on the full-body. We hypothesize that multiple clusters will emerge with a higher functional group, a lower functional group including mostly patients with bilateral knee OA, a limping group with upper body gait deviations. The level of

function will be assessed by comparison to with the control group in terms of walking speed, gait kinematics, and spatio-temporal parameters.

## 2 Materials and methods

### Participant's selection and characteristics

One hundred patients planned for unilateral primary TKA due to end-stage knee OA were included in this study, along with 33 healthy participants recruited as a control group. Thirty-nine patients had bilateral knee OA. Patients with previous lower limb arthroplasty, a history of lower limb or back surgery or neurologic or any other orthopaedic disorder that could affect gait or balance, were excluded. Only patients who performed a clinical gait analysis both before and 1 year after surgery, without any other joint arthroplasty in between, were considered. The healthy participants were matched in age and gender with the patient population and were included in the absence of symptomatic knee OA or other orthopaedic or neurologic problems potentially affecting gait or balance.

The database included patients from 2 consecutive, prospective protocols with the same inclusion/exclusion criteria applied, measured at the same gait lab. The first dataset included 79 the second included 21 patients. Patients in both datasets underwent primary TKA and had a a follow-up of measurement at 1 year (12.8 ± 0.6months, range between 10.4 and 16.3months). Both protocols were validated by the local ethics committee (CRE 09–307 and CCER 2018–00819). Patients were recruiteed between 01-03-2010 and 31-12-2012 for the first protocol (including control group) and between 01-09-2018 and 01-10-2021 for the second protocol. All patients and participants gave written informed consent.

The surgical procedure and postoperative management have been previously described [15]. The surgery were done routinely by a medial parapatellar approach, with use of fully cemented implants. For the first dataset, most of the patients had a tibia first technique, using a mechanical alignment. A fixed bearing was used in all patients but in the case of ultra-congruent (UC) constraint. In the second dataset, a kinematic alignment was preferred when using TKA with a medial pivot polyethylene, with a femur first, caliper-measurement technique. Cementing technique has been always realised in a two-step manner, by means of antibiotic loaded PMMA (Palacos). Patella resurfacing has been carried out routinely for the second dataset, independently from the constraint used, given a tendency towards too many revision for anterior knee pain.

Sex, age, pre- and post-operative Body Mass Index (BMI), localization of OA, and the physical status assessed by the American Society of Anaesthesiologists (ASA) score [16] were collected from our hospital-based arthroplasty registry (The Geneva Arthroplasty Registry) [17].

### Patient reported outcome measures

During their clinical pathways, patients filled the following patients reported outcome measures (PROMs): the Western Ontario and McMaster Universities Arthritis Index (WOMAC) before and after surgery [18, 19], the 12-item short form survey (SF-12) before and after surgery, and satisfaction with surgery one year after the surgery. The satisfaction was assessed with a Lickert scale questionnaire containing the following items: very dissatisfied, dissatisfied, neutral, satisfied, or very satisfied. In the present study, the score was dichotomized between satisfied (satisfied or very satisfied scores) and non-satisfied (very dissatisfied, dissatisfied, neutral). The level of pain during gait was assessed at the end of the clinical gait analysis with a Visual Analog Scale and was graded between 0 (no pain) and 10 (worst pain imaginable).

## Gait measurements

Patients performed a clinical gait analysis before and one year after surgery while the control group was evaluated only once. At each visit, participants were equipped with reflective skin markers according to the Conventional Gait Model 1.0 [20] and were asked to walk at self-selected speed on a 10m walkway. The markers trajectories were measured with 12 cameras opto-electronic systems (Vicon MX3 +, Oxford, UK for the first cohort and Oqus7+, Qualisys, Göteborg, Sweden for the second).

The kinematics of the segments and joints of the lower limb were computed according to the Conventional Gait Model 1.0 conventions [20] and the spatio-temporal parameters of gait were computed based on the events computed by the method of Fonseca et al. [21] and manually checked.

## Data selection and reduction

The included degrees of freedom (DoF) in the study were the three DoF of the thorax and pelvis, the hip flexion extension and adduction/abduction angle, the knee flexion and varus/valgus, the ankle flexion/extension and foot progression angle. The hip rotation angle was excluded from the analysis due to its poor reliability [22, 23] while the markerset did not allow measurements of the ankle inversion/eversion. The knee kinematics was excluded to focus on full-body deviations.

The kinematic time-series were reduced to key kinematics features summarising the kinematic profiles according to the method of Chehab et al. [24]. This method extracts the value at foot strike, the main peaks and the range of motion for each DoF. It was generalized for the thorax and foot progression angle.

To assess the multiple dimensions of gait, the following spatio-temporal parameters were included in the study: walking speed (m/s), stride time (s), stride length (m), cadence (steps/min), step time (s), step length (m), step width (m), time of foot off (% gait cycle), single support time (% of gait cycle) and the double support time (% of gait cycle).

The data used for this paper can be found with the following DOI: 10.26037/yareta:4xnhbsiyqrbihlep2nst2u2hyq.

## Data classification

The classification was performed with the K-means algorithm in Matlab (R2021a, The Math-Works, Inc., Natick, Massachusetts, United States) with the Squared Euclidean distance as a criterion and 30 replicates of unsupervised K-means++ initialization to ensure the convergence of the method. The silhouette criterion [25] was used to select the number of clusters. This widely used tool represents a measure of how well an object belongs to its cluster when compared to the closest cluster. By averaging the silhouette criterion of all objects, it is possible to have an estimation of how well the clusters are separated. The criterion was computed for 2 to 10 clusters and the number with the highest silhouette was selected as optimal and used in the study.

The classification was performed on the data of the pre-surgery clinical gait analysis. To focus on the full-body deviations, the knee kinematics features were excluded from the classification. The final dataset used in the clustering was a matrix including the 43 kinematics features and the 10 spatio-temporal parameters of the 100 patients. The control group was not included in the clustering.

## Data analysis

After data clustering, the clusters pre-surgery were compared to each other in terms of clinical data, kinematics features, and spatio-temporal data. The analysis was blind to the type of surgery or implant. The Kruskall-Wallis test with post-hoc Wilcoxon tests was used to compare the multiple clusters for all continuous data. The Chi2 test was used for proportions (satisfied or not, side of surgery, bilateral knee OA, sex, type of implant, dataset). The evolution of each cluster was assessed by comparing the clinical data, kinematics features, and spatio-temporal data measured pre-surgery to the ones measured one year after surgery with Wilcoxon tests. Finally, each cluster was compared before and after surgery to the control group with a Wilcoxon test. Statistical significance was set at $p < 0.05$.

## 3 Results

### Characteristics of clusters before surgery

The silhouette criterion was optimal for 3 clusters. The first one (CL1) had 59 patients (32F, median [IQR], 69.0 [8.8] years old, 168.0 [15.4] cm, BMI of 28.5 [5.x] kg/m$^2$), the second one (CL2) 20 patients (15F, 67.0 [13.5] years old, 161.8 [8.5] cm, BMI of 29.5 [7.5] kg/m$^2$) and the last one (CL3) 21 patients (16F, 74.0 [10.1] years old, 156.5 [16.1] cm, BMI of 31 [10.2] kg/m$^2$) (Table 1). There were no significant differences between groups in terms of sex, age, height, or BMI. However, CL3 was significantly older and shorter than the control group while all clusters had significantly larger BMI than the control group. CL1 had significantly less pain during gait than CL2 and CL3 but there were no other differences between clusters in terms of PROMs (Fig 1 and S1 Table). For all patients, the PROMs were lower than the levels of the control group. There was no significant difference in proportion of patients from each dataset in each cluster, respectively 15.3%, 30% and 29% of patients from cohort 2 for CL1, CL2 and CL3 and with the porporition for the whole cohort (21%). There were differences in terms of implant type between datasets but there was no differences between clusters (Table 2).

Regarding joint and segment kinematics pre-surgery, CL1 was mainly characterized by varus knee and sagittal plane pelvis and hip deviations compared the control group (Fig 1 and S2 Table). CL2 was characterized by gait deviations of the thorax, pelvis, and knee sagittal plane kinematics (Fig 1 and S2 Table). CL3 was characterized by deviations of the thorax and pelvis in frontal and transverse plane including increased RoM (e.g. thorax frontal) and

**Table 1. Clinical features and patient reported outcome measures of patients (median [IQR] or n patient & percentage), clusters and control group.**

| Features | All Patients (n = 100) | Cluster 1 (n = 59) | Cluster 2 (n = 20) | Cluster 3 (n = 21) | Control Group | KW or CHI2 | Between Clusters Comp. | Comp. with Control Group | | |
|---|---|---|---|---|---|---|---|---|---|---|
| | | | | | | | | CL1 | CL2 | CL3 |
| Sex | 63F (63%) | 32F (54%) | 15F (75%) | 16F (76%) | 18F (54%) | - | - | - | - | - |
| Age (years) | 69 [10.4] | 69 [8.8] | 67 [13.5] | 74 [10.1] | 66 [10.5] | - | - | - | - | < 0.01 |
| ASA [1–2] | 94% | 95% | 95% | 90% | - | - | - | - | - | - |
| Height (cm) | 162.3 [14.5] | 168 [15.4] | 161.8 [8.5] | 156.5 [16.1] | 166.5 [13.8] | - | - | - | - | 0.027 |
| BMI (kg/m2) | 29.1 [6.9] | 28.5 [5] | 29.5 [7.5] | 31 [10.2] | 24.2 [3.3] | - | - | < 0.01 | < 0.01 | < 0.01 |
| Side | 57R (57%) | 29R (49%) | 13R (65%) | 15R (71%) | - | - | - | - | - | - |
| Bilat | 39 (39%) | 26 (44%) | 6 (30%) | 7 (33%) | - | - | - | - | - | - |

KW stands for Kruskall-Walis tests between clusters (performed for continuous features) and CHI2 stands for the Chi-square test performed between clusters for proportion features. Between clusters comparison regroups the post-hoc tests (Wilcoxon or Chi2). Comp. with Control Group shows the comparison between each cluster and the control group (Wilcoxon or Chi2). Only the p-values below 0.05 were reported for clarity.

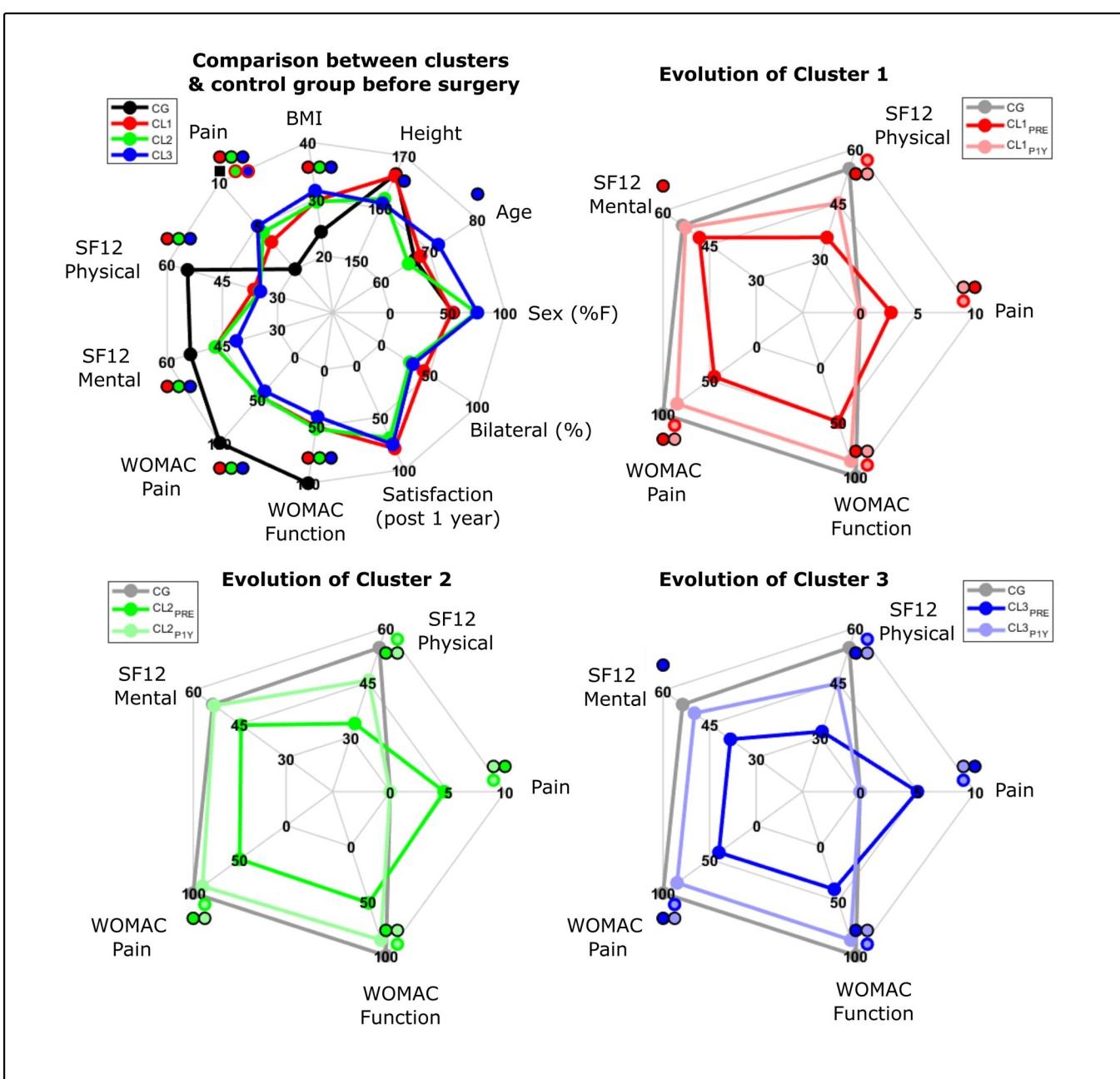

**Fig 1. Characteristics and patients reported outcomes for patient clusters before and one year after surgery compared to the control group.** Black square represents significant effect between clusters. Circles with two colors represent significant differences between the groups (CL1, CL2, CL3 or CG) represented by each colors.

modified patterns (pelvis frontal) as well as the external foot progression angles and reduced ranges and peak values of the hip, knee and ankle in the sagittal plane (Fig 1 and S2 Table).

Regarding the spatio-temporal parameters, CL1 and CL2 were similar when compared to the control group while CL3 was significantly poorer than the control group and the two other clusters (Fig 2 and S3 Table).

**Table 2. Implant characteristics of the whole cohort, datasets and clusters.**

| | All Patients (n = 100) | Dataset 1 (n = 79) | Dataset 2 (n = 21) | Between Cohort Comp. | Cluster 1 (n = 59) | Cluster 2 (n = 20) | Cluster 3 (n = 21) | Between Clusters Comp. |
|---|---|---|---|---|---|---|---|---|
| **Etiology** | | | | | | | | |
| Primary OA | 77% | 73% | 90% | - | 76% | 65% | 90% | - |
| Secondary OA | 23% | 27% | 10% | - | 24% | 35% | 10% | - |
| **Type of Implant** | | | | | | | | |
| GMK (PS/UC) | 20% | 25% | 0% | < 0.01 | 24% | 15% | 14% | - |
| GMK SPHERE | 12% | 0% | 57% | < 0.01 | 8% | 15% | 19% | - |
| PFC PS | 63% | 72% | 29% | < 0.01 | 63% | 65% | 62% | - |
| UKA (preservation/HP) | 2% | 3% | 0% | < 0.01 | 3% | 0% | 0% | - |
| PERSONA PS | 3% | 0% | 14% | < 0.01 | 2% | 5% | 5% | - |
| **Patellar resurfacing** | | | | | | | | |
| Yes (%) | 54% | 49% | 71% | < 0.01 | 49% | 65% | 57% | - |
| **Constraint** | | | | | | | | |
| PS | 77% | 87% | 38% | < 0.01 | 76% | 80% | 76% | - |
| Medial Congruent | 13% | 0% | 62% | < 0.01 | 8% | 20% | 19% | - |
| Ultra Congruent | 8% | 10% | 0% | < 0.01 | 12% | 0% | 5% | - |

Comp. stands for comparison of proportion performed with Chi-square between datasets and between clusters. Only the p-values below 0.05 were reported for clarity.

### Evolution of clusters one year after surgery

After one year, all groups improved all PROMs and did not show differences between them. However, CL3 showed the largest increase in terms of SF-12 mental scores. The WOMAC pain and function scores and SF-12 physical score remained below the control group level while the SF-12 mental score of patients reached the same level (Fig 1 and S1 Table).

Patients of CL1 presented multiple slight improvements in terms of kinematics and spatio-temporal parameters (Figs 3 and S1 and S4 Table). Patients in CL2 did not present any significant improvement (Figs 3 and S2 and S4 Table) while CL3 patients presented the largest improvements (Figs 3 and S3 and S4 Table). All groups remained below the control group levels and CL3 remained below CL1 and CL2.

## 4 Discussion

The aim of this study was to idenfity patterns of full-body gait deviations of patients with end-stage knee OA by performing an automatic classification based on full-body joint and segment kinematic profiles and spatio-temporal parameters evaluated during gait. The knee kinematics was purposefully removed from the classification to orient it toward the rest of the body.

Three clusters (CL1, CL2 and CL3) have been identified before surgery. They were first compared among each other and to a control group, and then, their evolution one year after TKA was assessed. The most affected cluster, in terms of preoperative functional limitations, was CL3 (n = 21). This cluster regrouped mostly elderly women, and was especially character-ised by slow walking speed (0.7 [0.2] m/s) and reduced range of motion in the sagittal plane (thorax, pelvis, hip, knee and ankle), by trunk and pelvis deviations (underlined by increased peaks in the frontal plane for the thorax and pelvis and by pelvis elevation during the stance phase), and, finally by externally rotated feet, typical of gait in the elderly [26]. "Lateral trunk lean toward the ipsilateral" side was reported by Iijima et al. as a typical characteristic of knee OA patients but was weakly correlated with pain [13]. Our study supports this result, but mostly for a subgroup of patients with the highest deficits of function but not necessarily the

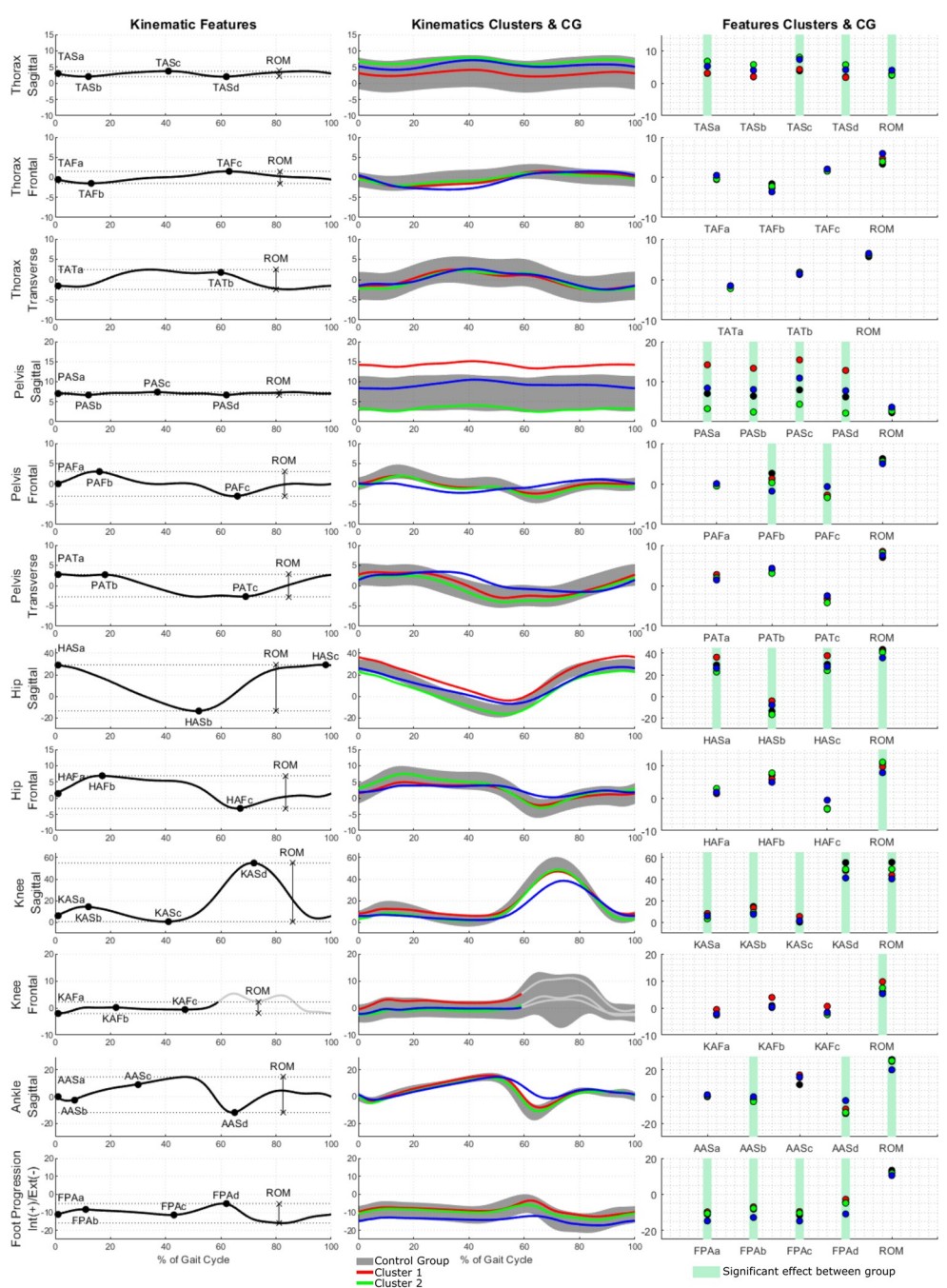

**Fig 2. Kinematics timeseries and features for patients clusters, with between clusters comparison.**

highest pain level. Indeed, CL3 had higher levels of pain with respect to CL1 but not with respect to the higher functioning group CL2. CL3 had slower gait speed than all other groups, however, as gait speed impacts mostly the amplitude of the gait parameters and not their pattern [27], this deviation of pattern is likely typical of this population. This low functioning cluster showed a tendency toward lower mental scores before surgery but there were no significant differences. One year after surgery, this cluster showed the largest functional improvement,

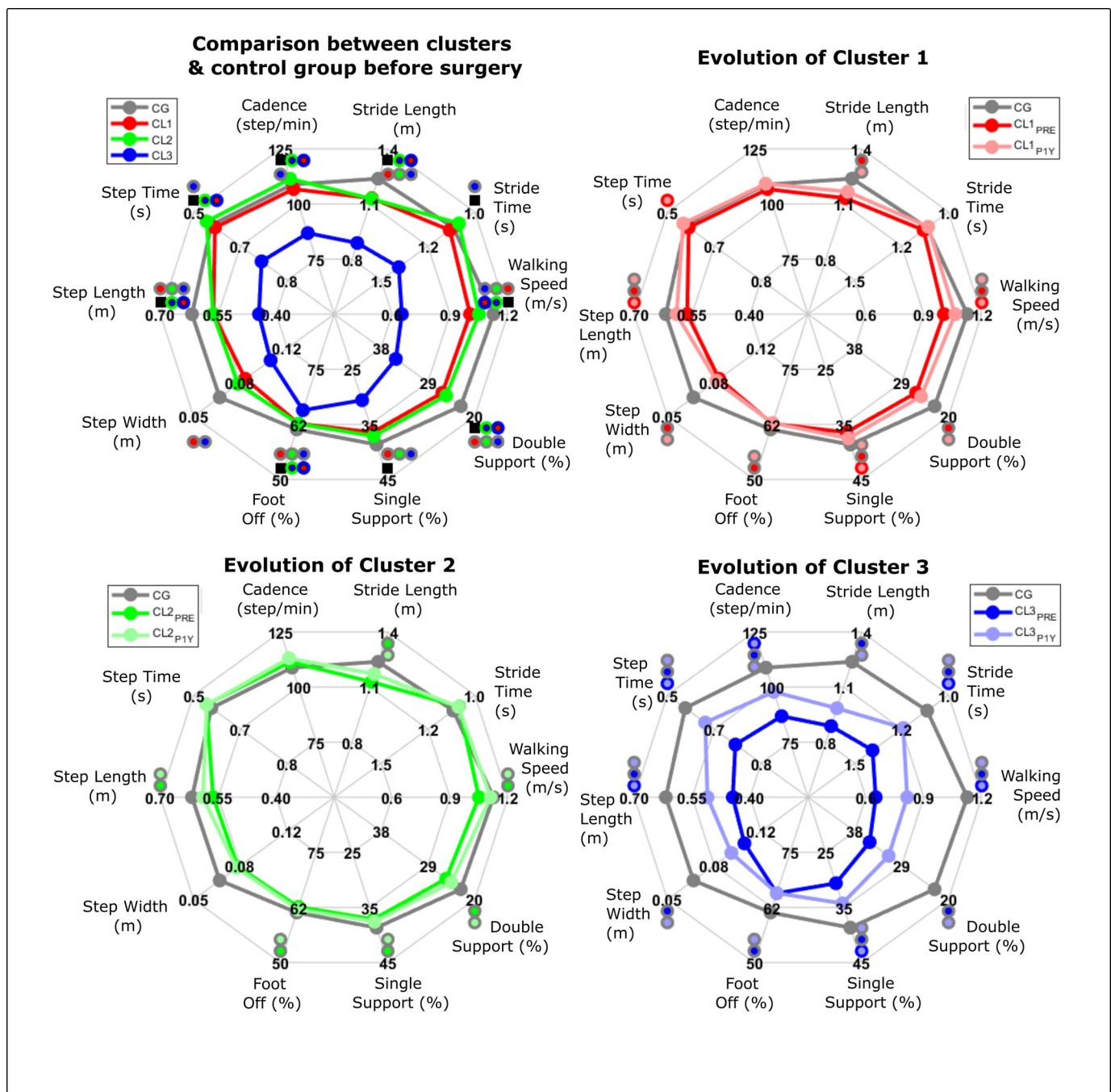

**Fig 3. Spatiotemporal parameters for patients clusters (before and one year after surgery) and control group.** Black square represents significant effect between clusters. Circles with two colors represent significant differences between the groups (CL1, CL2, CL3 or CG) represented by each colors.

but remained below both the level of the two other clusters and the control group. Indeed, despite improvement in spatio-temporal parameters, kinematic features and PROMs, most of the gait deviations identified before surgery were still present one year post-surgery. This cluster was similar to the "low function female cluster" identified among four clusters by Young-Shand et al. [8], while performing a classification based on knee kinematics, kinetics, and patient characteristics (age, sex, BMI). In this recent study, the low function women group

regrouped women with the slowest walking speed (mean [SD]: 0.7 [0.1] m/s), high BMI (35.1 (6.6) kg/m$^2$), knee flexum during gait, and it presented the largest functional improvement one year after surgery. However, this very group [8] was younger (mean [SD]: 62.9 [8.1] y) than the cluster CL3 (median [IQR]: 74.0 [10.1]) identified in our study.

CL1 (n = 59) and CL2 (n = 20) were similar in terms of patient characteristics (i.e. age, sex, BMI) and spatio-temporal parameters, but CL1 presented significantly less pain before surgery than CL2 and larger step width than the control group. Both clusters presented lower walking speed when compared to the control group, but the kinematic profiles were close to it, and differed mostly in terms of pelvis tilt (anteverted for CL1 and retroverted for CL2) and hip flexion, knee alignment (varus for CL1 and aligned for CL2), and knee flexion during stance phase (flexum for CL1 and fully extended for CL2). Although the knee kinematics was excluded in the clustering process, CL1 and CL2 still presented differences in this outcome. This results, reflects that, despite similar functional levels reflected by their similar walking speed, different knee phenotype (varus/neutral) may have an impact of the full-body mechanics of the patients.

Interestingly, CL1 had multiple improvements one year after surgery in terms of WOMAC pain and function, physical component of SF-12, spatiotemporal parameters, and kinematic features of gait, including a reduction of the varus, while CL2 improved only self-reported clinical outcomes scores and did not present changes in gait parameters one year after surgery. The only evolution of function in this cluster was seen in self-reported clinical outcomes, with a decrease in pain and improvements in WOMAC function, as well as the physical component of SF-12. Such discrepancy between objective and subjective scores post-surgery is well known in the literature [28] and can be explained by the pain reduction after surgery [29] that makes activities feel easier despite the lack of improvement in objective scores [28, 30] or even with lower function scores, as observed by Mizner et al. one month post-surgery [31].

The tendency toward retroverted pelvis and flexed thorax in CL2 indicates a posture typical of elderly patients that may be linked to sagittal imbalance [32]. These gait limitations can suggest lower functional abilities in CL2 than CL1, that may explain the post-operative evolution of this cluster. However, the tendency observed toward larger step width in CL1 before and 1 year after surgery highlights a "safe" walking pattern, priorizing balance over speed [33] and may indicate a balance deficit in CL1. The assessment of complementary functional domains such as muscle strength or balance would be necessary to clarify the differences between before and after surgery for these two clusters.

Overall, for CL2, the improvement after surgery was mainly in terms of pain reduction, while both pain score and functional outcomes improved for CL1 and especially CL3. These results are also in line with the study of Young-Shand et al. [8] that found groups of patients with higher and lower levels of function before surgery, with the lower function groups improving more their function post-surgery. Lower function groups have theoretically more range of improvement than higher function groups; however, higher function groups do not reach the level of control groups and still have room for improvement, too.

Interestingly, before surgery, CL1 presented a first peak of knee flexion, i.e the loading response showing weight acceptance and shock absorption after initial foot contact [34], while CL2 and CL3 do not present it. This absence of the first peak is typical of antalgic gait or of a quadriceps weakness [34] and differences may be explained by the higher pain scores of CL2 and CL3 with respect to CL1. Additionally, CL1 was characterized by valgus knee although the coronal deformity of the lower limb was not included in the clustering process. This suggests two points: 1) that valgus knee impacts the full-body kinematics and 2) that patients with a valgus knee may have lower functional deficits than the other two groups, as observed previously

in a subset of the present database [35] and despite having similar self-reported function scores as the other clusters [36].

One year after surgery, and despite differences in terms of kinematics and spatio-temporal parameters, there were no significant differences between clusters in terms of global satisfaction. Since satisfaction after surgery is linked to both pain reduction and improvement of function [37, 38], differences in functional outcomes were expected to lead to differences in satisfaction. Overall, 79% of the patients were satisfied, with 82%, 70% and 76% for CL1, CL2 and CL3 respectively. The satisfaction rate in CL2 was well below the 80% satisfaction rates reported one year after surgery at our institution [17] and in the literature [38]. Although not significant, this may indicate a link between the satisfaction and the absence of objective functional improvements after surgery. This lack of statistical significance may come from the number of patients in CL2 (n = 21) and CL3 (n = 20) that make it difficult to assess differences in percentage. Studies on larger cohorts may therefore confirm or infirm this trend. Since the main expectation before surgery, i.e. reduction of knee pain [39], was met in all clusters at one year, it could well be that the differences in function are less perceived and therefore do not impact strongly the satisfaction of patients. Nevertheless, all clusters improved their SF-12 mental scores well above the minimal important change of 1.4 identified in the literature [40] and reached the control group levels after surgery. This suggests a substantial impact of the surgery on the quality of life in all clusters. As for the satisfaction levels, this increase in mental score was observed in all clusters and thus may be linked to the decrease in pain more than the post-operative changes in function.

Contrary to our hypothesis, there was no differences in terms of unilateral or bilateral knee affection between the three clusters. These results supports studies by Creaby et al. [41] and Messier et al. [42] that agreed on the absence of differences in terms of lower limb kinematics between those two groups of patients, although their results differed on the symmetry parameters.

This study is not the first study using an unsupervised method to identify clusters of patients with severe KOA. However, comparison with existing literature is challenging because of the inherent diversity in motion capture systems, parameters included, patient profiles, and the selected classification algorithms and strategies. Petersen et al. [7] identified four clusters of OA knee function in a comprehensive analysis of the knee kinematics during gait with bi-plane fluoroscopy, the gold-standard method in motion capture. The optoelectronic system used in the present study cannot measure joint kinematics with such details [43] and it was chosen to focus on full-body kinematics, thus making it difficult to compare the consistency of these two clustering studies. In a study with the same modalities, Young-Shand et al. [8] identified four clusters separated by sex and level of function indicated by gait speed and knee flexum (high function male, high function female, low function male, low function female). Similarities in results with the present study were already discussed; however, our analysis may differ. Indeed, they suggest that high-function groups may not benefit so much from the surgery since there are no, or only slight, improvements in function one year after TKA. In the present study, the improvement in mental scores and satisfaction rates in higher function clusters (CL1 & CL2) underlines that pain reduction had positive effects on patients' quality of life after TKA as reported in the literature [44]. One can speculate that the intervention, although not improving objective functional outcomes for CL2, and only slightly for CL1, prevented a pejoration of patient's function, such as loss of muscle, joint stiffness, or consequent lower levels of activity due to the high levels of pain [45].

This study has some limitations. First, the sample size of the cohort. Although this cohort is among the largest in similar studies, a larger cohort could be more robust to define clusters and assess their differences, but, there is currently no consensus concerning the minimum

sample size of patients in such approaches. Secondly, the cohort combined datasets assessed with different motion capture systems brand and generations. A recent study from 2020 by Topley et al. [46] showed that modern systems were equivalent in terms of instrumental errors and key study from 2009 by Gorton et al. [47] showed that the instrumental error was way lower than marker misplacement errors when assessing variability. Thus, since both databases were measured in the same laboratory, with the same number of cameras, with the same marker set and same palpation techniques for marker placement, we assume that the instrumental errors between systems did not impact the results of the study. Third, the cohorts presented differences in terms of implant which was mostly due to the evolution of the implant market. However, the percentage of patients from each dataset was the same in each clusters as in the cohort and, there was no differences in terms of implant types between the clusters. This suggests that the post-surgery differences between clusters may be due to the pre-surgery functional profiles. The fact that the clustering was performed on pre-surgery data may have helped tempering the implants' influence. Fourth, a database from a single-center may limit between-patients variability and clustering. The replication of these results would help consolidate and validate this study, as previously done for clustering studies focusing on the pre-operative clinical profile of patients [48, 49]. Muscle force, pre- and postoperative lower limb alignment or type of implants (and their constraint) could have been relevant parameters in the determination of clusters, but were out of scope for the present study. Indeed, we chose to focus on full-body gait deviations since it was not clearly described in the literature. Soft tissue artefacts present in motion analysis approaches introduce errors into the resulting kinematic curves, especially outside the sagittal plane, as for hip or knee rotations, for instance, and do not allow the measurement of tibio-femoral displacements [43]. Moreover, to focus on full-body gait deviations and kinematics, we did not introduce the knee adduction-abduction moment, that is known to be key in patients with knee OA and TKA [4].

## 5 Conclusions

This study identified three functional groups of patients undergoing total knee arthroplasty, based on full-body kinematics during gait. One lower-functioning cluster regrouped mostly elderly women with the lowest function levels before and after surgery but with the most functional improvements after surgery. Two higher-functioning clusters separated mainly by pelvis tilt (anteversion vs. retroversion), knee alignment (varus vs. neutral), knee flexion during stance phase (flexum vs. extended) presented slight to no improvement after one year. Despite differences in terms of functional outcomes, the satisfaction rate was similar among clusters and mental scores improved for all clusters.

Higher functioning patients may benefit from TKA, mostly due to pain reduction, but may not see significant improvement of their function. On the contrary, patients with important functional limitation are more likely to improve both pain and functional outcomes. These point may help providing accurate information to patients to set realistic expectancies and improve patient satisfaction after surgery.

## Supporting information

**S1 Table. Patient reported outcome measures of patient clusters (before and one year after surgery) and of the control group.**
(DOCX)

**S2 Table. Kinematic features of patient clusters before surgery and comparison with control group.**
(DOCX)

**S3 Table. Spatio-temporal parameters of patient clusters (before and one year after surgery) and of the control group.**
(DOCX)

**S4 Table. Evolution of the kinematic features of patient clusters (before and one year after surgery) compared to the control group.**
(DOCX)

**S5 Table. Kinematic features for all patients before (PRE) and one year after surgery (P1Y) as well as all control group values.**
(CSV)

**S1 Fig. Kinematics timeries and features of cluster 1 before and one year after surgery.**
(PNG)

**S2 Fig. Kinematics timeries and features of cluster 2 before and one year after surgery.**
(PNG)

**S3 Fig. Kinematics timeries and features of cluster 3 before and one year after surgery.**
(PNG)

## Author Contributions

**Conceptualization:** Xavier Gasparutto, Alice Bonnefoy-Mazure, Stéphane Armand.

**Data curation:** Xavier Gasparutto, Alice Bonnefoy-Mazure, Michael Attias, Katia Turcot.

**Formal analysis:** Xavier Gasparutto, Alice Bonnefoy-Mazure, Stéphane Armand, Hermès H. Miozzari.

**Funding acquisition:** Hermès H. Miozzari.

**Investigation:** Xavier Gasparutto, Alice Bonnefoy-Mazure, Stéphane Armand, Hermès H. Miozzari.

**Methodology:** Xavier Gasparutto, Alice Bonnefoy-Mazure, Stéphane Armand.

**Project administration:** Hermès H. Miozzari.

**Software:** Xavier Gasparutto.

**Supervision:** Stéphane Armand, Hermès H. Miozzari.

**Writing – original draft:** Xavier Gasparutto.

**Writing – review & editing:** Alice Bonnefoy-Mazure, Michael Attias, Katia Turcot, Stéphane Armand, Hermès H. Miozzari.

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
