## [Decision Letter · Decision Letter 0]

29 Aug 2024

PONE-D-24-24263Comprehensive Analysis of Total Knee Arthroplasty Kinematics and Functional Recovery: Exploring Full-Body Compensations in Patients with Knee OsteoarthritisPLOS ONE

Dear Dr. Gasparutto,

Thank you for submitting your manuscript to PLOS ONE. After careful consideration, we feel that it has merit but does not fully meet PLOS ONE’s publication criteria as it currently stands. Therefore, we invite you to submit a revised version of the manuscript that addresses the points raised during the review process.

Some concerns were raised by the two reviewers. One concern is that the patients were analyzed at different times with different measurement systems and that there is a lack of information on the implants used and that both unilaterally and bilaterally affected patients were included in the study. These points should be addressed in more detail in the discussion and the limitations section.

Another issue that was raised was that the authors identify three patient groups according to their strategy to walk but have not clarifies how they know if these strategies are compensatory mechanisms. Could it be that these people just walked in a particular manner but that it is not a compensatory mechanism? While comparisons are made to the control group it is unclear whether clusters also exist in healthy patients and whether any of those clusters is similar to the TKA clusters.

A final major issue is that the authors hypothesize that three groups can be considered: high functional, low functional, limping group, but do not provide any criteria to be used to assess function. Without these criteria, it is unclear that the hypothesis can be tested. Also, no statistical analyses are presented to confirm that the three obtained groups have different kinematics and spatio-temporal parameters of gait.

We look forward to receiving your revised manuscript.

Kind regards,

John Leicester Williams, Ph.D.

Academic Editor

PLOS ONE

Journal requirements: 1. When submitting your revision, we need you to address these additional requirements. Please ensure that your manuscript meets PLOS ONE's style requirements, including those for file naming. The PLOS ONE style templates can be found at https://journals.plos.org/plosone/s/file?id=wjVg/PLOSOne_formatting_sample_main_body.pdf and https://journals.plos.org/plosone/s/file?id=ba62/PLOSOne_formatting_sample_title_authors_affiliations.pdf. 2. Thank you for stating the following financial disclosure:  [This study was funded by the "Fondation pour la recherche ostéoarticulaire" of Geneva (Switzerland).].  Please state what role the funders took in the study.  If the funders had no role, please state: ""The funders had no role in study design, data collection and analysis, decision to publish, or preparation of the manuscript."" If this statement is not correct you must amend it as needed. Please include this amended Role of Funder statement in your cover letter; we will change the online submission form on your behalf.  3. Thank you for stating the following in the Acknowledgments Section of your manuscript: [This work was supported by the Department of orthopaedic surgery and trauma care at the Geneva University Hospitals and by the “Fondation pour la recherche ostéoarticulaire” of Geneva. The funding source did not play a role in the design of this study.]We note that you have provided funding information that is not currently declared in your Funding Statement. However, funding information should not appear in the Acknowledgments section or other areas of your manuscript. We will only publish funding information present in the Funding Statement section of the online submission form. Please remove any funding-related text from the manuscript and let us know how you would like to update your Funding Statement. Currently, your Funding Statement reads as follows:  [This study was funded by the "Fondation pour la recherche ostéoarticulaire" of Geneva (Switzerland).] Please include your amended statements within your cover letter; we will change the online submission form on your behalf. 4. Please provide a complete Data Availability Statement in the submission form, ensuring you include all necessary access information or a reason for why you are unable to make your data freely accessible. If your research concerns only data provided within your submission, please write "All data are in the manuscript and/or supporting information files" as your Data Availability Statement.

Reviewers' comments:

Reviewer's Responses to Questions

**Comments to the Author**

1. Is the manuscript technically sound, and do the data support the conclusions?

Reviewer #1: Yes

Reviewer #2: Partly

2. Has the statistical analysis been performed appropriately and rigorously? 

Reviewer #1: Yes

Reviewer #2: No

3. Have the authors made all data underlying the findings in their manuscript fully available?

Reviewer #1: Yes

Reviewer #2: Yes

4. Is the manuscript presented in an intelligible fashion and written in standard English?

Reviewer #1: Yes

Reviewer #2: Yes

5. Review Comments to the Author

Reviewer #1: This is a solid study. An unsupervised approach has been chosen, to determine several

clusters from full-body gait kinematic and spatio-temporal parameters in patients with unilateral and bilateral knee osteoarthritis (OA). Three functional groups of patients undergoing total knee arthroplasty (TKA) were identified. One low-functioning cluster regrouped mostly elderly women with the lowest function levels before and after surgery but with the most functional improvements after surgery. Two high-functioning clusters separated mainly by pelvis tilt (anteversion vs. retroversion), knee alignment (varus vs. neutral), and knee flexion during stance phase (flexum vs. extended) presented slight to no improvement after one year. The authors concluded that high functioning patients may benefit from TKA, mostly due to pain reduction, but may not see significant improvement of their function. On the contrary, patients with important functional limitation are more likely to improve both pain and functional outcomes.

Although it is an interesting study approach, I have some concerns about the generalizability of the study results. My main concern is that the patients were analyzed at different times with different measurement systems. Secondly, there is a lack of information on the implants used and both unilaterally and bilaterally affected patients were included. These points should be addressed in more detail in the discussion and the limitations section.

The following are specific minor comments or recommendations:

ABSTRACT

• The authors stated that the knee was excluded from classification to focus on full-body kinematics (page 2, line 35). However, the results indicated two high-functioning clusters differentiated by pelvis tilt, sagittal knee alignment (varus vs. neutral), and knee flexion during stance phase (flexum vs. extended) (page 2, lines 38-42). Is this not a contradiction? Please also clarify/modify this point throughout the manuscript.

1 INTRODUCTION

• Page 5, lines 77-78: I suggest being more precise: “...the external knee adduction moment”…

• Page 5, lines 80-82: I suggest adding a few references and being more precise how knee OA impacts patients gait.

• Page 5 and 6, lines 96-98: I suggest an uniform definition of the objectives/hypotheses, both in the abstract and here at the end of the Introduction.

2 MATERIALS AND METHODS

Participant’s selection and characterstics

• How did the study group ensured that the reference group and the patient group (contralateral side for the unilateral affected patients) did not have asymptomatic knee osteoarthritis? X-rays were probably not possible for ethical reasons, were they?

• Page 6, line 113: …”and were performed at the same gait lab.” This information is redundant (see the first sentence of this paragraph).

• Page 6, lines 111-112: Please be more precise with the information regarding the follow-up measurement (e.g., mean and standard deviation / range).

• Page 6, line 117: Even though the authors list a reference here, it would be desirable for the reader to learn a little more about the implant used. So, were the same implants always used? Especially as the two measurement dates of the patients are very far apart (2010 – 2012 vs. 2018 – 2021). What effect could the type of implant have on the results?

Gait measurements

• The Conventional Gait Model 1.0 was implemented in 2019. Please clarify which gait model (marker placement) was used for the first cohort (2010 – 2012) and discuss potential differences regarding the gait model. Please also discuss potential differences by using two motion capture systems (VICON vs. Qualisys).

Data Classification

• Page 8, line 162: The authors stated that “the optimal number of clusters was identified with the silhouette criterion.” Please specify and add a reference.

Data Analysis

• Page 9, lines 171-172: The authors stated that “the Chi2 test was used for proportions (satisfied or not, side of surgery, bilateral knee OA, sex).” Please clarify, if patients with bilateral knee OA were included as well!? Up to this point, I thought that only unilateral patients had been included. Please discuss the potential effect of including bilateral affected patients besides unilateral affected patients! Even if the difference in the distribution of unilateral and bilateral patients is not different between the groups, in my opinion, it would have been better to include only unilaterally affected patients in the analysis.

3 RESULTS

Characteristics of clusters before surgery

• Table 1: p-values for the sex distribution are missing. Please also add the superior p-values for KW or CHI2 and not only the p-values for the pairwise comparison.

4 DISCUSSION

• Significant differences in walking speed exists between patient clusters (2 vs. 3 and 3 vs. 1) and compared to the control group (all clusters). Please discuss the potential effect of the difference in walking speed on the results of the study and conclusions made.

Reviewer #2: In this study, the authors utilize an unsupervised clustering method to identify mechanisms that candidates for total knee arthroplasty (TKA) may utilize to compensate for pain or lack of function and to determine whether these mechanisms persist after TKA. To this end, the authors analyze a series of whole-body kinematic parameters (excluding the knee) and spatiotemporal parameters of gait. The authors then compare these groups among them and to a control group as a function of their patient reported outcomes. The authors identify three phenotypes from the TKA patients. Identifying different phenotypes of TKA patients is a crucial step to understand how the surgery can be tailored to these groups to improve success and satisfaction. In this way, the manuscript is interesting, because it investigates how other joints may compensate for lack of optimal knee function. However, I have some concerns with the study in its current form, which dampen my enthusiasm for the presented work. Some comments are, in order:

The goal of the manuscript is to identify clusters of TKA candidates by their compensatory mechanisms. The authors identify three groups according to their strategy to walk. However, how can the authors be sure that these are compensatory mechanisms? Could it be that these people just walked in a particular manner but that it is not a compensatory mechanism? In this way, the authors compare the groups to the control group; however, it is unclear whether clusters also exist in healthy patients and whether any of those clusters is similar to the TKA clusters.

The authors hypothesize that three groups would appear: high functional, low functional, limping group. However, the authors do not provide any criteria for which criteria will be used to assess function (i.e., is it a particular gait or spatiotemporal parameter or is it a compound of various metrics?) or any criteria for establishing whether a patient has high or low function. Without these criteria, it is unclear that the hypothesis can be tested. More importantly, the authors do not present statistical analysis that confirms that the three obtained groups have different kinematics and spatiotemporal parameters of gait. In this way, it is unclear from the results whether the hypothesis can be confirmed or rejected.

I find the way the results and discussion are presented confused. The authors compare groups among themselves at the same time that they compare them to the control group. For example, in lines 256-258, it is unclear whether the TKA clusters were different in their pain and step width or was just CL1 compared to healthy. To achieve their goal of identifying patterns of full body compensation, I would argue that the authors have to show that differences exist in the kinematics across TKA patients (other than the knee), that these differences are caused by compensatory mechanisms, how these differences relate to knee kinematics, and how these differences affect outcomes (i.e., PROMS). I believe most of this is in the manuscript, but it is not presented in a manner that helps identify/highlight these differences, making it hard for the reader to understand the message of the manuscript. I suggest re-structuring the results to present the clusters, discussing their demographics. Then, present their differences in compensatory mechanisms (i.e., why were they identified as different groups?). Then present their differences in knee kinematics and relate these to the control group. Finally, discuss their differences in PROMs and relate these to surgical technique or implant/limb alignment.

Other comments/questions:

How was lack of knee OA assessed in the healthy group?

Were there differences in surgical technique or implant position among patients in the different clusters?

L105 – did the participants had any other joint affected by advanced OA, which may have contributed to their affected gait? Was this different across groups?

L 110 – what were these inclusion/exclusion criteria?

L117 – can the authors include a brief description of the surgical procedure? While the authors provide a reference, the

L162 – can the authors briefly describe the silhouette criterion?

L184-185 – The authors indicate in the methods that the control group was matched by age/sex/bmi to the TKA patients. How is it possible that one of the clusters is significantly older than the controls?

L200-206 –can the authors be more specific as to the direction of the deviations?

L217-225 – PROMs are one of the main outcomes evaluated and used in the discussion; however, they are included in a supplement and not in the main manuscript. Can the authors please include the PROMs scores and their changes and whether these changes reached the MCID?

L276 – how did the authors conclude that the pattern is “safe”? What makes a pattern “safe”?

L258-261 – can the authors quantify how “close” the parameters were to the control group? Perhaps determining whether they were statistically significant/clinically meaningful or not.

L262 – can the authors describe the improvements in WOMAC? Were these above MCID?

6. PLOS authors have the option to publish the peer review history of their article (what does this mean?). If published, this will include your full peer review and any attached files.

Reviewer #1: No

Reviewer #2: No

---

## [Author Response · Author response to Decision Letter 0]

16 Oct 2024

Response to reviewers was uploaded as a separate file.

---

## [Editor Report · Decision Letter 1]

20 Nov 2024

Comprehensive Analysis of Total Knee Arthroplasty Kinematics and Functional Recovery: Exploring Full-Body Compensations in Patients with Knee Osteoarthritis

PONE-D-24-24263R1

Dear Dr. Gasparutto,

We’re pleased to inform you that your manuscript has been judged scientifically suitable for publication and will be formally accepted for publication once it meets all outstanding technical requirements.

Kind regards,

John Leicester Williams, Ph.D.

Academic Editor

PLOS ONE
---

## [Editor Report · Acceptance letter]

22 Nov 2024

PONE-D-24-24263R1 

PLOS ONE

Dear Dr. Gasparutto, 

I'm pleased to inform you that your manuscript has been deemed suitable for publication in PLOS ONE. Congratulations! Your manuscript is now being handed over to our production team.

Kind regards, 

on behalf of

Dr. John Leicester Williams 

Academic Editor

PLOS ONE